# RSN: Randomized Subspace Newton

**Robert M. Gower**
LTCI, Télécom Paristech, IPP, France
gowerrobert@gmail.com

**Dmitry Kovalev**
KAUST, Saudi Arabia
dmitry.kovalev@kaust.edu.sa

**Felix Lieder**
Heinrich-Heine-Universität Düsseldorf, Germany
lieder@opt.uni-duesseldorf.de

**Peter Richtárik**
KAUST, Saudi Arabia and MIPT, Russia
peter.richtarik@kaust.edu.sa

## Abstract

We develop a randomized Newton method capable of solving learning problems with huge dimensional feature spaces, which is a common setting in applications such as medical imaging, genomics and seismology. Our method leverages randomized sketching in a new way, by finding the Newton direction constrained to the space spanned by a random sketch. We develop a simple global linear convergence theory that holds for practically all sketching techniques, which gives the practitioners the freedom to design custom sketching approaches suitable for particular applications. We perform numerical experiments which demonstrate the efficiency of our method as compared to accelerated gradient descent and the full Newton method. Our method can be seen as a refinement and randomized extension of the results of Karimireddy, Stich, and Jaggi [18].

## 1  Introduction

In this paper we are interested in unconstrained optimization problems of the form

$$\min_{x \in \mathbb{R}^d} f(x), \tag{1}$$

where $f : \mathbb{R}^d \to \mathbb{R}$ is a sufficiently well behaved function, in the *large dimensional* setting, i.e., when $d$ is very large. Large dimensional optimization problems are becoming ever more common in applications. Indeed, $d$ often stands for the dimensionality of captured data, and due to fast-paced advances in technology, this only keeps growing. One of key driving forces behind this is the rapid increase in the resolution of sensors used in medicine [19], genomics [26, 8], seismology [2] and weather forecasting [1]. To make predictions using such high dimensional data, typically one needs to solve an optimization problem such as (1). The traditional off-the-shelf solvers for such problems are based on Newton's method, but in this large dimensional setting they cannot be applied due to the high memory footprint and computational costs of solving the Newton system. We offer a new solution to this, by iteratively performing Newton steps in random subspaces of sufficiently low dimensions. The resulting randomized Newton's method need only solve small randomly compressed Newton systems and can be applied to solving (1) no matter how big the dimension $d$.

### 1.1  Background and contributions

Newton's method dates back to even before Newton, making an earlier appearance in the work of the Persian astronomer and mathematician al-Kashi 1427 in his "Key to Arithmetic" [33]. In the 80's Newton's method became the workhorse of nonlinear optimization methods such as trust region [9], augmented Lagrangian [4] and interior point methods. The research into interior point methods

culminated with Nesterov and Nemirovskii's [22] ground breaking work proving that minimizing a convex (self-concordant) function could be done in a polynomial number of steps, where in each step a Newton system was solved.

Amongst the properties that make Newton type methods so attractive is that they are invariant to rescaling and coordinate transformations. This property makes them particularly appealing for off-the-shelf solvers since they work well independently of how the user chooses to scale or represent the variables. This in turn means that Newton based methods need little or no tuning of hyperparameters. This is in contrast with first-order methods[1], where even rescaling the function can result in a significantly different sequence of iterates, and their efficient execution relies on parameter tuning (typically the stepsize).

Despite these advantages, Newton based solvers are now facing a challenge that renders most of them inapplicable: large dimensional feature spaces. Indeed, solving a generic Newton system costs $O(d^3)$. While inexact Newton methods [11, 5] made significant headway to diminishing this high cost by relying on Krylov based solvers whose iterations cost $O(d^2)$, this too can be prohibitive, and this is why first order methods such as accelerated gradient descent [24] are often used in the large dimensional setting.

In this work we develop a family of randomized Newton methods which work by leveraging randomized sketching and projecting [16]. The resulting randomized Newton method has a **global linear convergence** for virtually any type and size of sketching matrix. In particular, one can choose a sketch of size one, which yields a **low iteration complexity** of as little as $O(1)$ if one assumes that scalar derivatives can be computed in constant time. Our main assumptions are the recently introduced [18] **relative smoothness and convexity**[2] of $f$, which are in a certain sense weaker than the more common strong convexity and smoothness assumptions. Our method is also **scale invariant**, which facilitates setting the stepsize. We further propose an efficient line search strategy that does not increase the iteration complexity.

There are only a handful of Newton type methods in the literature that use iterative sketching, including the sketched Newton algorithm [28], SDNA (Stochastic Dual Newton Ascent) [29], RBCN (Randomized Block Cubic Newton) [12] and SON [21]. In the unconstrained case the sketched Newton algorithm [28] requires a sketching matrix that is proportional to the global rank of the Hessian, an unknown constant related to high probability statements and $\epsilon^{-2}$, where $\epsilon > 0$ is the desired tolerance. Consequently, the required sketch size could be as large as $d$, which defeats the purpose.

The SDNA algorithm in [29] relies on the existence of a positive definite matrix $\mathbf{M} \in \mathbb{R}^{d \times d}$ that globally upper bounds the Hessian, which is a stronger assumption than our relative smoothness assumption. The method then proceeds by selecting random principal submatrices of $\mathbf{M}$ that it then uses to form and solve an approximate Newton system. The theory in [29] allows for any sketch size, including size of one. Our method could be seen as an extension of SDNA to allow for any sketch, one that is directly applied to the Hessian (as opposed to $\mathbf{M}$) and one that relies on a set of more relaxed assumptions. The RBCN method combines the ideas of randomized coordinate descent [23] and cubic regularization [25]. The method requires the optimization problem to be block separable and is hence not applicable to the problem we consider here. Finally, SON [21] uses random and deterministic streaming sketches to scale up a second-order method, akin to a Gauss–Newton method, for solving online learning problems.

## 1.2 Key Assumptions

We assume throughout that $f : \mathbb{R}^d \to \mathbb{R}$ is a convex and twice differentiable function. Further, we assume that $f$ is bounded below and the set of minimizers $\mathcal{X}_*$ nonempty. We denote the optimal value of (1) by $f_* \in \mathbb{R}$.

Let $\mathbf{H}(x) := \nabla^2 f(x)$ (resp. $g(x) = \nabla f(x)$) be the Hessian (resp. gradient) of $f$ at $x$. We fix an initial iterate $x_0 \in \mathbb{R}^d$ throughout and define $\mathcal{Q}$ to be a level set of function $f(x)$ associated with $x_0$:

$$\mathcal{Q} := \left\{ x \in \mathbb{R}^d : f(x) \leq f(x_0) \right\}. \tag{2}$$

Let $\langle x, y \rangle_{\mathbf{H}(x_k)} := \langle \mathbf{H}(x_k)x, y \rangle$ for all $x, y \in \mathbb{R}^d$. Our main assumption on $f$ is given next.

**Assumption 1.** *There exist constants $\hat{L} \geq \hat{\mu} > 0$ such that for all $x, y \in \mathcal{Q}$:*

$$f(x) \leq \underbrace{f(y) + \langle g(y), x - y \rangle + \tfrac{\hat{L}}{2}\|x - y\|^2_{\mathbf{H}(y)}}_{:=T(x,y)}, \tag{3}$$

$$f(x) \geq f(y) + \langle g(y), x - y \rangle + \tfrac{\hat{\mu}}{2}\|x - y\|^2_{\mathbf{H}(y)}. \tag{4}$$

*We refer to $\hat{L}$ and $\hat{\mu}$ as the* relative smoothness *and* relative convexity *constant, respectively.*

Relative smoothness and convexity is a direct consequence of smoothness and strong convexity. It is also a consequence of the recently introduced [18] $c$–stability condition, which served to us as an inspiration. Specifically, as shown in Lemma 2 in [18] and also formally (for convenience) stated in Proposition 2 in the supplementary material, we have that

$$L\text{–smooth} + \mu\text{–strongly convex} \quad \Rightarrow \quad c\text{–stability} \quad \Rightarrow \quad \text{relative smoothness \& relative convexity.}$$

We will also further assume:

**Assumption 2.** $g(x) \in \text{Range}\left(\mathbf{H}(x)\right)$ *for all $x \in \mathbb{R}^d$.*

Assumption 2 holds if the Hessian is positive definite for all $x$, and for generalized linear models.

### 1.3 The full Newton method

Our baseline method for solving (1), is the following variant of the Newton Method (NM):

$$x_{k+1} \;=\; x_k + \gamma n(x_k) \;:=\; x_k - \gamma \mathbf{H}^\dagger(x_k)g(x_k), \tag{5}$$

where $\mathbf{H}^\dagger(x_k)$ is the Moore-Penrose pseudoinverse of $\mathbf{H}(x_k)$ and $n(x_k) := -\mathbf{H}^\dagger(x_k)g(x_k)$ is the Newton direction. A property (which we recall from [18]) that will be important for our analysis is that for a suitable stepsize, Newton's method is a descent method.

**Lemma 1.** *Consider the iterates $\{x_k\}_{k\geq 0}$ defined recursively by (5). If $\gamma \leq 1/\hat{L}$ and (3) holds, then $f(x_{k+1}) \leq f(x_k)$ for all $k \geq 0$, and in particular, $x_k \in \mathcal{Q}$ for all $k \geq 0$.*

The proof follows by using (3), twice differentiability and convexity of $f$. See [18, Lemma 3].

The relative smoothness assumption (3) is particularily important for motivating Newton's method. Indeed, a Newton step is the exact minimizer of the upper bound in (3).

**Lemma 2.** *If Assumption 2 is satisfied, then the quadratic $x \mapsto T(x, x_k)$ defined in (3) has a global minimizer $x_{k+1}$ given by $x_{k+1} = x_k - \frac{1}{\hat{L}}\mathbf{H}^\dagger(x_k)g(x_k) \in \mathcal{Q}$.*

*Proof.* Lemma 1 implies that $x_{k+1} \in \mathcal{Q}$, and Lemma 9 in the appendix shows that (5) is a global minimizer for $\gamma = 1/\hat{L}$. □

## 2 Randomized Subspace Newton

Solving a Newton system exactly is costly and may be a waste of resources. Indeed, this is the reason for the existence of inexact variants of Newton methods [11]. For these inexact Newton methods, an accurate solution is only needed when close to the optimal point.

In this work we introduce a different inexactness idea: we propose to solve an *exact* Newton system, but in an *inexact* randomly selected subspace. In other words, we propose a *randomized subspace Newton method*, where the randomness is introduced via *sketching matrices*, defined next.

**Definition 1.** Let $\mathcal{D}$ be a (discrete or continuous) distribution over matrices in $\mathbb{R}^{d \times s}$. We say that $\mathbf{S} \sim \mathcal{D}$ is a random sketching matrix and $s \in \mathbb{N}$ is the sketch size.

We will often assume that the random sketching is *nullspace preserving*.

**Assumption 3.** *We say that $\mathbf{S} \sim \mathcal{D}$ is* nullspace preserving *if with probability one we have that*

$$\text{Null}\left(\mathbf{S}^\top \mathbf{H}(x)\mathbf{S}\right) = \text{Null}(\mathbf{S}), \qquad \forall x \in \mathcal{Q}. \tag{6}$$

---
**Algorithm 1** RSN: Randomized Subspace Newton
---
1: **input:** $x_0 \in \mathbb{R}^d$
2: **parameters:** $\mathcal{D}$ = distribution over random matrices
3: **for** $k = 0, 1, 2, \dots$ **do**
4:     sample a fresh sketching matrix: $\mathbf{S}_k \sim \mathcal{D}$
5:     $x_{k+1} = x_k - \frac{1}{\hat{L}} \mathbf{S}_k \left( \mathbf{S}_k^\top \mathbf{H}(x_k) \mathbf{S}_k \right)^\dagger \mathbf{S}_k^\top g(x_k)$
6: **output:** last iterate $x_k$
---

By sampling a sketching matrix $\mathbf{S}_k \sim \mathcal{D}$ in the $k$th iteration, we can form a *sketched Newton* direction using only the sketched Hessian $\mathbf{S}_k^\top \mathbf{H}(x_k) \mathbf{S}_k \in \mathbb{R}^{s \times s}$; see line 5 in Algorithm 1. Note that the sketched Hessian is the result of twice differentiating the function $\lambda \mapsto f(x_k + \mathbf{S}_k \lambda)$, which can be done efficiently using a single backpropation pass [14] or $s$ backpropagation passes [7] which costs at most $s$ times the cost of evaluating the function $f$.

First we show that much like the full Newton method (5), Algorithm 1 is a descent method.

**Lemma 3** (Descent). *Consider the iterates $x_k$ given Algorithm 1. If Assumptions 1, 2 and 3 hold, then $f(x_{k+1}) \leq f(x_k)$ and consequently $x_k \in \mathcal{Q}$ for all $k \geq 0$.*

While common in the literature of randomized coordinate (subspace) descent method, this is a rare result for randomized stochastic gradient descent methods, which do not enjoy a descent property. Lemma 3 is useful in monitoring the progress of the method in cases when function evaluations are not too prohibitive. However, we use it solely for establishing a tighter convergence theory.

Interestingly, the iterations of Algorithm 1 can be equivalently formulated as a random projection of the full Newton step, as we detail next.

**Lemma 4.** *Let Assumptions 1 and 2 hold. Consider the projection matrix $\mathbf{P}_k$ with respect to the seminorm $\|\cdot\|_{\mathbf{H}(x_k)}^2 := \langle \cdot, \cdot \rangle_{\mathbf{H}(x_k)}$ given by*

$$\mathbf{P}_k := \mathbf{S}_k \left( \mathbf{S}_k^\top \mathbf{H}(x_k) \mathbf{S}_k \right)^\dagger \mathbf{S}_k^\top \mathbf{H}(x_k) \in \mathbb{R}^{d \times d}. \tag{7}$$

*The iterates of Algorithm 1 can be viewed as a projection of the Newton step given by*

$$x_{k+1} = x_k + \frac{1}{\hat{L}} \mathbf{P}_k n(x_k). \tag{8}$$

*Proof.* To verify that $\mathbf{P}_k$ is an oblique projection matrix, it suffices to check that

$$\langle \mathbf{P}_k x, \mathbf{P}_k y \rangle_{\mathbf{H}(x_k)} = \langle \mathbf{P}_k x, y \rangle_{\mathbf{H}(x_k)}, \quad \forall x, y \in \mathbb{R}^d,$$

which in turn relies on the identity $\mathbf{M}^\dagger \mathbf{M} \mathbf{M}^\dagger = \mathbf{M}^\dagger$, which holds for all matrices $\mathbf{M} \in \mathbb{R}^{d \times d}$. Since $g(x_k) \in \text{Range}(\mathbf{H}(x_k))$, we have again by the same identity of the pseudoinverse that

$$g(x_k) = \mathbf{H}(x_k) \mathbf{H}^\dagger(x_k) g(x_k) = -\mathbf{H}(x_k) n(x_k). \tag{9}$$

Consequently, $\mathbf{P}_k n(x_k) = \mathbf{S}_k \left( \mathbf{S}_k^\top \mathbf{H}(x_k) \mathbf{S}_k \right)^\dagger \mathbf{S}_k^\top g(x^k)$.  $\square$

We will refer to $\mathbf{P}_k n(x_k)$ as the *sketched Newton direction*. If we add one more simple assumption to the selection of the sketching matrices, we have the following equivalent formulations of the sketched Newton direction.

**Lemma 5.** *Let Assumptions 1, 2 and 3 hold. It follows that the $x_{k+1}$ iterate of Algorithm 1 can be equivalently seen as*

*1. The minimizer of $T(x, x_k)$ over the random subspace $x \in x_k + \text{Range}(\mathbf{S}_k)$ :*

$$x_{k+1} = x_k + \mathbf{S}_k \lambda_k, \quad where \quad \lambda_k \in \arg\min_{\lambda \in \mathbb{R}^s} T(x_k + \mathbf{S}_k \lambda, x_k). \tag{10}$$

*Furthermore,*

$$T(x_{k+1}, x_k) = f(x_k) - \frac{1}{2\hat{L}} \|g(x_k)\|_{\mathbf{S}_k (\mathbf{S}_k^\top \mathbf{H}(x_k) \mathbf{S}_k)^\dagger \mathbf{S}_k}^2. \tag{11}$$

*2. A projection of the Newton direction onto a random subspace:*

$$x_{k+1} \;=\; \arg\min_{x\in\mathbb{R}^d,\,\lambda\in\mathbb{R}^s}\left\|x-\left(x_k-\tfrac{1}{\hat{L}}n(x_k)\right)\right\|_{\mathbf{H}(x_k)}^2 \quad \text{subject to} \quad x=x_k+\mathbf{S}_k\lambda. \quad (12)$$

*3. A projection of the previous iterate onto the* sketched *Newton system given by:*

$$x_{k+1} \;\in\; \arg\min\|x-x_k\|_{\mathbf{H}(x_k)}^2 \quad \text{subject to} \quad \mathbf{S}_k^\top\mathbf{H}(x_k)(x-x_k)=-\tfrac{1}{\hat{L}}\mathbf{S}_k^\top g(x_k). \quad (13)$$

*Furthermore, if* $\mathrm{Range}\,(\mathbf{S}_k)\subset\mathrm{Range}\,(\mathbf{H}_k(x_k))$, *then* $x_{k+1}$ *is the unique solution to the above.*

## 3 Convergence Theory

We now present two main convergence theorems.

**Theorem 2.** *Let* $\mathbf{G}(x) \coloneqq \mathbb{E}_{\mathbf{S}\sim\mathcal{D}}\left[\mathbf{S}\left(\mathbf{S}^\top\mathbf{H}(x)\mathbf{S}\right)^\dagger\mathbf{S}\right]$ *and define*

$$\rho(x) \coloneqq \min_{v\in\mathrm{Range}(\mathbf{H}(x))}\frac{\langle\mathbf{H}^{1/2}(x)\mathbf{G}(x)\mathbf{H}^{1/2}(x)v,v\rangle}{\|v\|_2^2} \qquad \text{and} \qquad \rho \coloneqq \min_{x\in\mathcal{Q}}\rho(x). \quad (14)$$

*If Assumptions 1 and 2 hold, then*

$$\mathbb{E}\left[f(x_k)\right]-f_* \leq \left(1-\rho\tfrac{\hat{\mu}}{\hat{L}}\right)^k\left(f(x_0)-f_*\right). \quad (15)$$

*Consequently, given* $\epsilon>0$, *if* $\rho>0$ *and if*

$$k\geq\tfrac{1}{\rho}\tfrac{\hat{L}}{\hat{\mu}}\log\left(\tfrac{f(x_0)-f_*}{\epsilon}\right), \qquad \text{then} \qquad \mathbb{E}\left[f(x_k)-f_*\right]<\epsilon. \quad (16)$$

Theorem 2 includes the convergence of the full Newton method as a special case. Indeed, when we choose[3] $\mathbf{S}_k=\mathbf{I}\in\mathbb{R}^{d\times d}$, it is not hard to show that $\rho(x_k)\equiv 1$, and thus (16) recovers the $\hat{L}/\hat{\mu}\log(1/\epsilon)$ complexity given in [18]. We provide yet an additional sublinear $\mathcal{O}(1/k)$ convergence result that holds even when $\hat{\mu}=0$.

**Theorem 3.** *Let Assumption 2 hold and Assumption 1 be satisfied with* $\hat{L}>\hat{\mu}=0$. *If*

$$\mathcal{R} \coloneqq \inf_{x_*\in\mathcal{X}_*}\sup_{x\in\mathcal{Q}}\|x-x_*\|_{\mathbf{H}(x)}<+\infty\,, \quad (17)$$

*and* $\rho>0$ *then* $\mathbb{E}\left[f(x_k)\right]-f_*\leq\frac{2\hat{L}\mathcal{R}^2}{\rho k}$.

As a new result of Theorem 3, we can also show that the full Newton method has a $O(\hat{L}\mathcal{R}\epsilon^{-1})$ iteration complexity.

Both of the above theorems rely on $\rho>0$. So in the next Section 3.1 we give sufficient conditions for $\rho>0$ that holds for virtually all sketching matrices.

### 3.1 The sketched condition number $\rho(x_k)$

The parameters $\rho(x_k)$ and $\rho$ in Theorem 2 characterize the trade-off between the cost of the iterations and the convergence rate of RSN. Here we show that $\rho$ is always bounded between one and zero, and further, we give conditions under which $\rho(x_k)$ is the smallest *non-zero* eigenvalue of an expected projection matrix, and is thus bounded away from zero.

**Lemma 6.** *The parameter* $\rho(x_k)$ *appearing in Theorem 2 satisfies* $0\leq\rho(x_k)\leq 1$. *Letting*

$$\hat{\mathbf{P}}(x_k) \coloneqq \mathbf{H}^{1/2}(x_k)\mathbf{S}_k\left(\mathbf{S}_k^\top\mathbf{H}(x_k)\mathbf{S}_k\right)^\dagger\mathbf{S}_k^\top\mathbf{H}^{1/2}(x_k)\,, \quad (18)$$

*and if we assume that the* exactness[4] *condition*

$$\mathrm{Range}\,(\mathbf{H}(x_k))=\mathrm{Range}\left(\mathbb{E}_{\mathbf{S}\sim\mathcal{D}}\left[\hat{\mathbf{P}}(x_k)\right]\right) \quad (19)$$

*holds then* $\rho(x_k)=\lambda_{\min}^+\left(\mathbb{E}_{\mathbf{S}\sim\mathcal{D}}\left[\hat{\mathbf{P}}(x_k)\right]\right)>0$.

Since (19) is in general hard to verify, we give simpler sufficient conditions for $\rho > 0$ in the next lemma.

**Lemma 7** (Sufficient condition for exactness). *If Assumption 3 and*

$$\text{Range}\left(\mathbf{H}(x_k)\right) \subset \text{Range}\left(\text{E}[\mathbf{S}_k\mathbf{S}_k^\top]\right), \tag{20}$$

*holds then* (19) *holds and consequently* $0 < \rho \leq 1$.

Clearly, condition (20) is immediately satisfied if $\mathbb{E}\left[\mathbf{S}_k\mathbf{S}_k^\top\right]$ is invertible, and this is the case for Gaussian sketches, weighted coordinate sketched, sub-sampled Hadamard or Fourier transforms, and the entire class of randomized orthonormal system sketches [27].

## 3.2 The relative smoothness and strong convexity constants

In the next lemma we give an insightful formula for calculating the relative smoothness and convexity constants defined in Assumption 1, and in particular, show how $\hat{L}$ and $\hat{\mu}$ depend on the *relative change* of the Hessian.

**Lemma 8.** *Let $f$ be twice differentiable, satisfying Assumption 1. If moreover $\mathbf{H}(x)$ is invertible for every $x \in \mathbb{R}^d$, then*

$$\hat{L} = \max_{x,\,y\,\in\,\mathcal{Q}} \int_{t=0}^1 2(1-t)\frac{\|z_t-y\|_{\mathbf{H}(z_t)}^2}{\|z_t-y\|_{\mathbf{H}(y)}^2}dt \leq \max_{x,y\in\mathcal{Q}}\frac{\|x-y\|_{\mathbf{H}(x)}^2}{\|x-y\|_{\mathbf{H}(y)}^2} := c \tag{21}$$

$$\hat{\mu} = \min_{x,\,y\,\in\,\mathcal{Q}} \int_{t=0}^1 2(1-t)\frac{\|z_t-y\|_{\mathbf{H}(z_t)}^2}{\|z_t-y\|_{\mathbf{H}(y)}^2}dt \geq \tfrac{1}{c}, \tag{22}$$

*where $z_t := y + t(x - y)$.*

The constant $c$ on the right hand side of (21) is known as the $c$-stability constant [18]. As a by-product, the above lemma establishes that the rates for the deterministic Newton method obtained as a special case of our general theorems are at least as good as those obtained in [18] using $c$-stability.

# 4 Examples

With the freedom of choosing the sketch size, we can consider the extreme case $s = 1$, i.e., the case with the sketching matrices having only a single column.

**Corollary 1** (Single column sketches). *Let $0 \prec \mathbf{U} \in \mathbb{R}^{n \times n}$ be a symmetric positive definite matrix such that $\mathbf{H}(x) \preceq \mathbf{U}, \; \forall x \in \mathbb{R}^d$. Let $\mathbf{D} = [d_1, \ldots, d_n] \in \mathbb{R}^{n \times n}$ be a given invertible matrix such that $d_i^\top \mathbf{H}(x)d_i \neq 0$ for all $x \in \mathcal{Q}$ and $i = 1, \ldots, n$. If we sample according to*

$$\mathbb{P}[\mathbf{S}_k = d_i] = p_i := \frac{d_i^\top \mathbf{U}d_i}{\text{Trace}(\mathbf{D}^\top\mathbf{U}\mathbf{D})},$$

*then the update on line 5 of Algorithm 1 is given by*

$$x_{k+1} = x_k - \frac{1}{\hat{L}}\frac{d_i^\top g(x_k)}{d_i^\top \mathbf{H}(x_k)d_i}\,d_i, \quad \text{with probability } p_i, \tag{23}$$

*and under the assumptions of Theorem 2, Algorithm 1 converges according to*

$$\mathbb{E}\left[f(x_k)\right] - f_* \leq \left(1 - \min_{x\in\mathcal{Q}}\frac{\lambda_{\min}^+(\mathbf{H}^{1/2}(x)\mathbf{D}\mathbf{D}^\top\mathbf{H}^{1/2}(x))}{\text{Trace}(\mathbf{D}^\top\mathbf{U}\mathbf{D})}\frac{\hat{\mu}}{\hat{L}}\right)^k (f(x_0) - f_*). \tag{24}$$

Each iteration of single colum sketching Newton method (23) requires only three scalar derivatives of the function $t \mapsto f(x_k + td_k)$ and thus if $f(x)$ can be evaluated in constant time, this amounts to $O(1)$ cost per iteration. Indeed (23) is much like coordinate descent, except we descent along the $d_i$ directions, and with a stepsize that adapts depending on the curvature information $d_i^\top \mathbf{H}(x_k)d_i$.[5]

The rate of convergence in (24) suggests that we should choose $\mathbf{D} \approx \mathbf{U}^{-1/2}$ so that $\rho$ is large. If there is no efficient way to approximate $\mathbf{U}^{-1/2}$, then the simple choice of $\mathbf{D} = \mathbf{I}$ gives $\rho(x_k) = \lambda_{\min}^+(\mathbf{H}(x_k))/\text{Trace}(\mathbf{U})$.

An expressive family of functions that satisfy Assumption 1 are *generalized linear models*.

**Definition 4.** Let $0 \leq u \leq \ell$. Let $\phi_i : \mathbb{R} \mapsto \mathbb{R}_+$ be a twice differentiable function such that

$$u \leq \phi_i''(t) \leq \ell, \quad \text{for } i = 1, \ldots, n. \tag{25}$$

Let $a_i \in \mathbb{R}^d$ for $i = 1, \ldots, n$ and $\mathbf{A} = [a_1, \ldots, a_n] \in \mathbb{R}^{d \times n}$. We say that $f : \mathbb{R}^d \to \mathbb{R}$ is a generalized linear model when

$$f(x) = \frac{1}{n} \sum_{i=1}^{n} \phi(a_i^\top x) + \frac{\lambda}{2} \|x\|_2^2 . \tag{26}$$

The structure of the Hessian of a generalized linear model is such that highly efficient fast Johnson-Lindenstrauss sketches [3] can be used. Indeed, the Hessian is given by

$$\mathbf{H}(x) \quad = \quad \frac{1}{n} \sum_{i=1}^{n} a_i a_i^\top \phi_i''(a_i^\top x) + \lambda \mathbf{I} = \frac{1}{n} \mathbf{A} \Phi''(\mathbf{A}^\top x) \mathbf{A}^\top + \lambda \mathbf{I} ,$$

and consequently, for computing the sketch Hessian $\mathbf{S}_k^\top \mathbf{H}(x_k) \mathbf{S}_k$ we only need to sketch the fixed matrix $\mathbf{S}_k^\top \mathbf{A}$ and compute $\mathbf{S}_k^\top \mathbf{S}_k$ efficiently, and thus no backpropgation is required. This is exactly the setting where fast Johnson–Lindenstrauss transforms can be effective [17, 3].

We now give a simple expression for computing the relative smoothness and convexity constant for generalized linear models.

**Proposition 1.** *Let $f : \mathbb{R}^d \to \mathbb{R}$ be a generalized linear model with $0 \leq u \leq \ell$. Then Assumption 1 is satisfied with*

$$\hat{L} = \frac{\ell \sigma_{\max}^2(\mathbf{A}) + n\lambda}{u \sigma_{\max}^2(\mathbf{A}) + n\lambda} \qquad and \qquad \hat{\mu} = \frac{u \sigma_{\max}^2(\mathbf{A}) + n\lambda}{\ell \sigma_{\max}^2(\mathbf{A}) + n\lambda}. \tag{27}$$

*Furthermore, if we apply Algorithm 1 with a sketch such that $\mathbb{E}\left[\mathbf{SS}^\top\right]$ is invertible, then the iteration complexity (16) of applying Algorithm 1 is given by*

$$k \geq \frac{1}{\rho} \left( \frac{\ell \sigma_{\max}^2(\mathbf{A}) + n\lambda}{u \sigma_{\max}^2(\mathbf{A}) + n\lambda} \right)^2 \log \left( \frac{1}{\epsilon} \right). \tag{28}$$

This complexity estimate (28) should be contrasted with that of gradient descent. When $x_0 \in \text{Range}(\mathbf{A})$, the iteration complexity of GD (gradient descent) applied to a smooth generalized linear model is given by $\frac{\ell \sigma_{\max}^2(\mathbf{A}) + n\lambda}{u \sigma_{\min+}^2(\mathbf{A}) + n\lambda} \log \left( \frac{1}{\epsilon} \right)$, where $\sigma_{\min+}(\mathbf{A})$ is the smallest non-zero singular value of $\mathbf{A}$. To simplify the discussion, and as a santiy check, consider the full Newton method with $\mathbf{S}_k = \mathbf{I}$ for all $k$, and consequently $\rho = 1$. In view of (28) *Newton method does not depend on the smallest singular values nor the condition number of the data matrix.* This suggests that for ill-conditioned problems Newton method can be superior to gradient descent, as is well known.

## 5   Experiments and Heuristics

In this section we evaluate and compare the computational performance of RSN (Algorithm 1) on generalized linear models (26). Specifically, we focus on logistic regression, i.e., $\phi_i(t) = \ln(1 + e^{-y_i t})$, where $y_i \in \{-1, 1\}$ are the target values for $i = 1, \ldots, n$. Gradient descent (GD), accelerated gradient descent (AGD) [24] and full Newton methods[6] are compared with RSN.

Table 1: Details of the data sets taken from LIBSM [6] and OpenML [31].

| dataset | non-zero features (d) | samples (n) | density |
|---|---|---|---|
| chemotherapy | 61,359 + 1 | 158 +1 | 1 |
| gisette | 5,000 + 1 | 6000 | 0.9910 |
| news20 | 1,355,191 + 1 | 19996 | 0.0003 |
| rcv1 | 47,237 + 1 | 20,241 | 0.0016 |
| real-sim | 20,958 + 1 | 72,309 | 0.0025 |
| webspam | 680,715 + 1 | 350,000 | 0.0055 |

For simplicity, block coordinate sketches are used; these are random sketch matrices of the form $\mathbf{S}_k \in \{0, 1\}^{d \times s}$ with exactly one non-zero entry per row and per column. We will refer to $s \in \mathbb{N}$ as the *sketch size*. To ensure fairness and for comparability purposes, all methods were supplied with

the exact Lipschitz constants and equipped with the same line-search strategy (see Algorithm 3 in the supplementary material). We consider 6 datasets with a diverse number of features and samples (see Table 1 for details) which were modified by removing all zero features and adding an intercept, i.e., a constant feature.

For regularization we used $\lambda = 10^{-10}$ and stopped methods once the gradients norm was below $tol = 10^{-6}$ or some maximal number of iterations had been exhausted. In Figures 1 to 3 we plotted iterations and wall-clock time vs gradient norm, respectively.

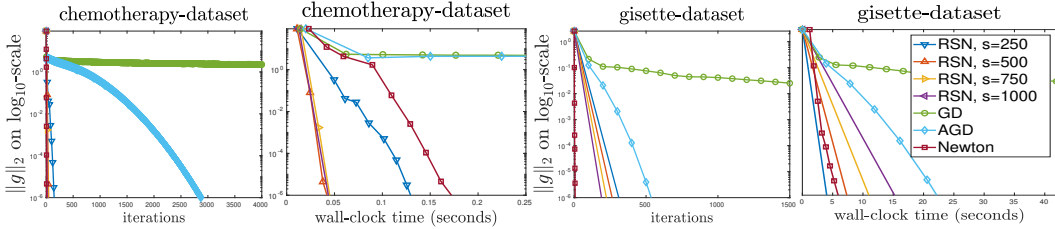

Figure 1: Highly dense problems, favoring RSN methods.

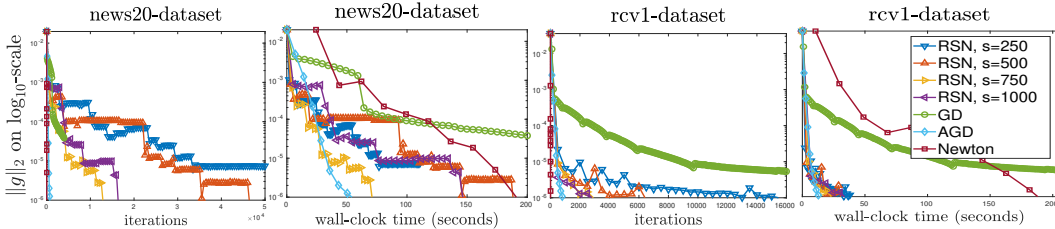

Figure 2: Due to extreme sparsity, accelerated gradient is competitive with the Newton type methods.

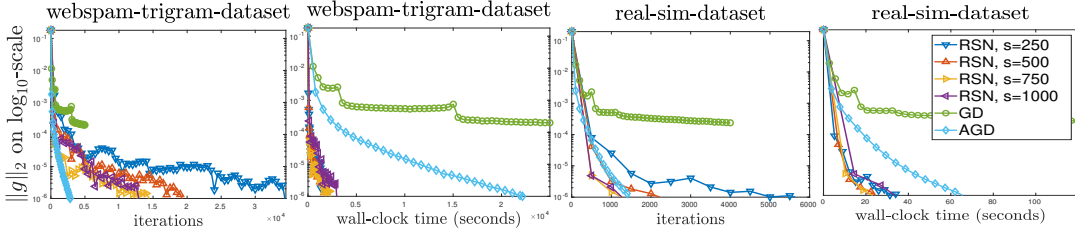

Figure 3: Moderately sparse problems favor the RSN method. The full Newton method is infeasible due to high dimensionality.

Newton's method, when not limited by the immense costs of forming and solving linear systems, is competitive as we can see in the gisette problem in Figure 1. In most real-world applications however, the bottleneck is exactly within the linear systems which may, even if they can be formed at all, require significant solving time. On the other end of the spectrum, GD and AGD need usually more iterations and therefore may suffer from expensive full gradient evaluations, for example due to higher density of the data matrix, see Figure 3. RSN seems like a good compromise here: As the sketch size and type can be controlled by the user, the involved linear systems can be kept reasonably sized. As a result, the RSN is the fastest method in all the above experiments, with the exception of the extremely sparse problem news20 in Figure 2, where AGD outruns RSN with $s = 750$ by approximately 20 seconds.

## 6   Conclusions and Future Work

We have laid out the foundational theory of a class of randomized Newton methods, and also performed numerical experiments validating the methods. There are now several venues of work to explore including 1) combining the randomized Newton method with subsampling so that it can be applied to data that is both high dimensional and abundant 2) leveraging the potential fast Johnson-Lindenstrauss sketches to design even faster variants of RSN 3) develop heuristic sketches based on past descent directions inspired on the quasi-Newton methods [15].

## Footnotes

[1]An exception to this is, for instance, the optimal first order affine-invariant method in [10].

[2]These notions are different from the *relative smoothness and convexity* concepts considered in [20].

[3]Or when $\mathbf{S}_k$ is an invertible matrix.

[4]An "exactness" condition similar to (19) was introduced in [30] in a program of "exactly" reformulating a linear system into a stochastic optimization problem. Our condition has a similar meaning, but we do not elaborate on this as this is not central to the developments in this paper.

[5]There in fact exists a block coordinate method that also incorporates second order information [13].

[6]To implement the Newton's method efficiently, of course we exploit the Sherman–Morrison–Woodbury matrix identity [32] when appropriate

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
