[Supplementary Material]

# Supplementary Material: Randomized Subspace Newton Method

## A  Key Lemmas

**Lemma 9.** *Let $y \in \mathbb{R}^d$, $c > 0$ and $\mathbf{H} \in \mathbb{R}^{d \times d}$ be a symmetric positive semi-definite matrix. Let $g \in \mathrm{Range}\,(\mathbf{H})$. The set of solutions to*

$$\hat{x} \in \arg\min_{x \in \mathbb{R}^d} \langle g, x - y \rangle + \frac{c}{2} \|x - y\|_{\mathbf{H}}^2, \tag{29}$$

*is given by*

$$\hat{x} \in \mathbf{H}^{\dagger} \left( \mathbf{H}y - \frac{1}{c}g \right) + \mathrm{Null}(\mathbf{H}). \tag{30}$$

*Two particular solutions in the above set are given by*

$$\hat{x} = y - \frac{1}{c}\mathbf{H}^{\dagger}g, \tag{31}$$

*and the least norm solution*

$$x^{\dagger} = \mathbf{H}^{\dagger} \left( \mathbf{H}y - \frac{1}{c}g \right). \tag{32}$$

*The minimum of* (29) *is*

$$\langle g, \hat{x} - y \rangle + \frac{c}{2} \|\hat{x} - y\|_{\mathbf{H}}^2 = -\frac{1}{2c} \|g\|_{\mathbf{H}^{\dagger}}^2. \tag{33}$$

*Proof.* Taking the derivative in $x$ and setting to zero gives

$$\frac{1}{c}g + \mathbf{H}(x - y) = 0.$$

The above linear system is guaranteed to have a solution because $g \in \mathrm{Range}(\mathbf{H})$. The solution set to this linear system is the set

$$\mathbf{H}^{\dagger}(\mathbf{H}y - \tfrac{1}{c}g) + \mathrm{Null}(\mathbf{H}).$$

The point (31) belong to the above set by noting that $(\mathbf{I} - \mathbf{H}^{\dagger}\mathbf{H})y \in \mathrm{Null}(\mathbf{H})$, which in turn follows by the $\mathbf{H} = \mathbf{H}\mathbf{H}^{\dagger}\mathbf{H}$ property of pseudoinverse matrices. Clearly (32) is the least norm solution.

Finally, using any solution (30) we have that

$$\hat{x} - y \in (\mathbf{H}^{\dagger}\mathbf{H} - \mathbf{I})y - \frac{1}{c}\mathbf{H}^{\dagger}g + \mathrm{Null}(\mathbf{H}),$$

which when substituted into (29) gives

$$(29) \quad = \quad \underbrace{\left\langle g, (\mathbf{H}^{\dagger}\mathbf{H} - \mathbf{I})y - \tfrac{1}{c}\mathbf{H}^{\dagger}g \right\rangle}_{\alpha} + \underbrace{\frac{c}{2} \left\| (\mathbf{H}^{\dagger}\mathbf{H} - \mathbf{I})y - \tfrac{1}{c}\mathbf{H}^{\dagger}g \right\|_{\mathbf{H}}^2}_{\beta}. \tag{34}$$

Since $g \in \mathrm{Range}\,(\mathbf{H})$ we have that $g^{\top}(\mathbf{H}^{\dagger}\mathbf{H} - \mathbf{I}) = 0$ and thus $\alpha = -\frac{1}{c}\|g\|_{\mathbf{H}^{\dagger}}^2$. Furthermore

$$\begin{aligned} \beta &= \left\| (\mathbf{H}^{\dagger}\mathbf{H} - \mathbf{I})y - \tfrac{1}{c}\mathbf{H}^{\dagger}g \right\|_{\mathbf{H}}^2 \\ &= \left\| (\mathbf{H}^{\dagger}\mathbf{H} - \mathbf{I})y \right\|_{\mathbf{H}}^2 - \frac{2}{c} \left\langle \mathbf{H}(\mathbf{H}^{\dagger}\mathbf{H} - \mathbf{I})y, \mathbf{H}^{\dagger}g \right\rangle + \frac{1}{c^2} \left\| \mathbf{H}^{\dagger}g \right\|_{\mathbf{H}}^2 \\ &= \frac{1}{c^2} \left\| \mathbf{H}^{\dagger}g \right\|_{\mathbf{H}}^2 \quad = \quad \frac{1}{c^2} \left\langle g, \mathbf{H}^{\dagger}\mathbf{H}\mathbf{H}^{\dagger}g \right\rangle \quad = \quad \frac{1}{c^2} \|g\|_{\mathbf{H}^{\dagger}}^2, \end{aligned}$$

where we used that $\mathbf{H}^{\dagger}\mathbf{H}\mathbf{H}^{\dagger} = \mathbf{H}^{\dagger}$. Using the above calculations in (34) gives

$$(29) \quad = \quad -\frac{1}{c}\|g\|_{\mathbf{H}^{\dagger}}^2 + \frac{1}{2c}\|g\|_{\mathbf{H}^{\dagger}}^2 \quad = \quad -\frac{1}{2c}\|g\|_{\mathbf{H}^{\dagger}}^2.$$

$\square$

**Lemma 10.** *For any matrix $\mathbf{W}$ and symmetric positive semidefinite matrix $\mathbf{G}$ such that*

$$\mathrm{Null}(\mathbf{G}) \subset \mathrm{Null}(\mathbf{W}^\top), \tag{35}$$

*we have that*

$$\mathrm{Null}(\mathbf{W}) = \mathrm{Null}(\mathbf{W}^\top \mathbf{G} \mathbf{W}) \tag{36}$$

*and*

$$\mathrm{Range}(\mathbf{W}^\top) = \mathrm{Range}(\mathbf{W}^\top \mathbf{G} \mathbf{W}). \tag{37}$$

*Proof.* In order to establish (36), it suffices to show the inclusion $\mathrm{Null}(\mathbf{W}) \supseteq \mathrm{Null}(\mathbf{W}^\top \mathbf{G} \mathbf{W})$ since the reverse inclusion trivially holds. Letting $s \in \mathrm{Null}(\mathbf{W}^\top \mathbf{G} \mathbf{W})$, we see that $\|\mathbf{G}^{1/2}\mathbf{W}s\|^2 = 0$, which implies $\mathbf{G}^{1/2}\mathbf{W}s = 0$. Consequently

$$\mathbf{W}s \in \mathrm{Null}(\mathbf{G}^{1/2}) = \mathrm{Null}(\mathbf{G}) \overset{(35)}{\subset} \mathrm{Null}(\mathbf{W}^\top).$$

Thus $\mathbf{W}s \in \mathrm{Null}(\mathbf{W}^\top) \cap \mathrm{Range}(\mathbf{W})$ which are orthogonal complements which shows that $\mathbf{W}s = 0$.

Finally, (37) follows from (36) by taking orthogonal complements. Indeed, $\mathrm{Range}(\mathbf{W}^\top)$ is the orthogonal complement of $\mathrm{Null}(\mathbf{W})$ and $\mathrm{Range}(\mathbf{W}^\top \mathbf{G} \mathbf{W})$ is the orthogonal complement of $\mathrm{Null}(\mathbf{W}^\top \mathbf{G} \mathbf{W})$. $\qquad\square$

Our assumptions are inspired on the $c$–stability assumption in [18]:

**Proposition 2** ([18] c-stable)**.** *We say that $f$ is c–stable if for every $y, z \in \mathcal{Q}$, $z \neq y$ we have that $\|z - y\|^2_{\mathbf{H}(y)} > 0$, and there exists a constant $c \geq 1$ such that*

$$c = \max_{y,z \in \mathcal{Q}} \frac{\|z - y\|^2_{\mathbf{H}(z)}}{\|z - y\|^2_{\mathbf{H}(y)}}. \tag{38}$$

*We say that $f$ is L–smooth if*

$$f(x) \leq f(y) + \langle g(y), x - y \rangle + \frac{L}{2}\|x - y\|^2_2, \tag{39}$$

*and $\mu$–strongly convex if*

$$f(x) \geq f(y) + \langle g(y), x - y \rangle + \frac{\mu}{2}\|x - y\|^2_2. \tag{40}$$

*If $f$ is $\mu$–strongly convex and L–smooth, then $f$ is $L/\mu$–stable. Furthermore if $f$ is c–stable then Assumption 1 holds with $\hat{L} \leq c$ and $\hat{\mu} \geq \frac{1}{c}$.*

*Proof.* Lemma 2 in [18] proves that $c$–stability implies $c$ relative smoothness and $c$ relative convexity. The inequalities $\hat{L} \leq c$ and $\hat{\mu} \geq \frac{1}{c}$ follow from (38) compared to (22) and (21). $\qquad\square$

## B   Proof of Lemma 2

*Proof.* Lemma 1 implies that $x_{k+1} \in \mathcal{Q}$, and Lemma 9 in the appendix shows that (5) is a global minimizer for $\gamma = 1/\hat{L}$. $\qquad\square$

## C   Proof of Lemma 3

*Proof.* Due to (10) we have that

$$f(x_{k+1}) \overset{(3)}{\leq} T(x_k, x_{k+1}) = \min_{\lambda \in \mathbb{R}^s} T(x_k, x_k + \lambda \mathbf{S}_k) \leq T(x_k, x_k) = f(x_k).$$

$\qquad\square$

# D Proof of Lemma 5

*Proof.*    1. Plugging in $y = x_k$ and $x = x_k + \mathbf{S}_k\lambda$ into (3) we have that

$$
\begin{aligned}
T(x_k + \mathbf{S}_k\lambda, x_k) &= f(x_k) + \langle g(x_k), \mathbf{S}_k\lambda \rangle + \frac{\hat{L}}{2}\|\mathbf{S}_k\lambda\|^2_{\mathbf{H}(y)} \\
&= f(x_k) + \langle \mathbf{S}_k^\top g(x_k), \lambda \rangle + \frac{\hat{L}}{2}\|\lambda\|^2_{\mathbf{S}_k^\top \mathbf{H}(x_k)\mathbf{S}_k}. \quad (41)
\end{aligned}
$$

By taking the orthogonal components in (6) we have that $\mathbf{S}_k^\top g(x_k) \in$ Range $\left(\mathbf{S}_k^\top \mathbf{H}(x_k)\mathbf{S}_k\right)$, and consequently from Lemma 9 we have that the minimizer is given by

$$
\lambda_k \in -\frac{1}{\hat{L}}\left(\mathbf{S}_k^\top \mathbf{H}(x_k)\mathbf{S}_k\right)^\dagger \mathbf{S}_k^\top g(x_k) + \text{Null}\left(\mathbf{S}_k^\top \mathbf{H}(x_k)\mathbf{S}_k\right). \quad (42)
$$

Left multiplying by $\mathbf{S}_k^\top$ gives

$$
\begin{aligned}
\mathbf{S}_k\lambda_k &= -\frac{1}{\hat{L}}\mathbf{S}_k\left(\mathbf{S}_k^\top \mathbf{H}(x_k)\mathbf{S}_k\right)^\dagger \mathbf{S}_k^\top g(x_k) + \mathbf{S}_k \text{Null}\left(\mathbf{S}_k^\top \mathbf{H}(x_k)\mathbf{S}_k\right) \\
&\overset{(6)}{=} -\frac{1}{\hat{L}}\mathbf{S}_k\left(\mathbf{S}_k^\top \mathbf{H}(x_k)\mathbf{S}_k\right)^\dagger \mathbf{S}_k^\top g(x_k) \\
&\overset{Lemma\ 4}{=} \frac{1}{\hat{L}}\mathbf{P}_k n(x_k). \quad (43)
\end{aligned}
$$

Consequently $x_k + \mathbf{S}_k\lambda_k = x_k + \frac{1}{\hat{L}}\mathbf{P}_k n(x_k)$.

Furthermore, since $\lambda_k$ is the minimizer of (41), we have from Lemma 9 and (33) that

$$
\begin{aligned}
T(x_{k+1}, x_k) &= T(x_k + \mathbf{S}_k\lambda_k) = f(x_k) - \frac{1}{2\hat{L}}\left\|\mathbf{S}_k^\top g(x_k)\right\|^2_{(\mathbf{S}_k^\top \mathbf{H}(x_k)\mathbf{S}_k)^\dagger} \\
&= f(x_k) - \frac{1}{2\hat{L}}\|g(x_k)\|^2_{\mathbf{S}_k(\mathbf{S}_k^\top \mathbf{H}(x_k)\mathbf{S}_k)^\dagger \mathbf{S}_k^\top}.
\end{aligned}
$$

2. Plugging in the constraint into the objective in (12) gives

$$
\begin{aligned}
\left\|\mathbf{S}_k\lambda + \frac{1}{\hat{L}}n(x_k)\right\|^2_{\mathbf{H}(x_k)} &= \|\lambda\|^2_{\mathbf{S}_k^\top \mathbf{H}(x_k)\mathbf{S}_k} + \frac{2}{\hat{L}}\left\langle \mathbf{S}_k^\top \mathbf{H}(x_k)n(x_k), \lambda \right\rangle + \frac{1}{\hat{L}^2}\|n(x_k)\|^2_{\mathbf{H}(x_k)} \\
&\overset{(9)}{=} \|\lambda\|^2_{\mathbf{S}_k^\top \mathbf{H}(x_k)\mathbf{S}_k} + \frac{2}{\hat{L}}\left\langle \mathbf{S}_k^\top g(x_k), \lambda \right\rangle + \frac{1}{\hat{L}^2}\|n(x_k)\|^2_{\mathbf{H}(x_k)}.
\end{aligned}
$$

Consequently minimizing the above is equivalent to minimizing (41), and thus $\mathbf{S}_k\lambda$ is given by (43).

3. The Lagrangian of (13) is

$$
L(d, \lambda) = \|x - x_k\|^2_{\mathbf{H}(x_k)} + \left\langle \lambda, \mathbf{S}_k^\top \mathbf{H}(x_k)(x - x_k) + \frac{1}{\hat{L}}\mathbf{S}_k^\top g(x_k) \right\rangle.
$$

Differentiating in $d$ and setting to zero gives

$$
\mathbf{H}(x_k)(x - x_k) + \mathbf{H}(x_k)\mathbf{S}_k\lambda = 0. \quad (44)
$$

Left multiplying by $\mathbf{S}_k^\top$ and using the constraint in (13) gives

$$
\mathbf{S}_k^\top \mathbf{H}(x_k)\mathbf{S}_k\lambda = \frac{1}{\hat{L}}\mathbf{S}_k^\top g(x_k). \quad (45)
$$

Again we have that $\mathbf{S}_k^\top g(x_k) \in \text{Range}\left(\mathbf{S}_k^\top \mathbf{H}(x_k)\mathbf{S}_k\right)$ by (6). Consequently by Lemma 9 we have that the solution set in $\lambda$ is given by

$$
\lambda = \frac{1}{\hat{L}}\left(\mathbf{S}_k^\top \mathbf{H}(x_k)\mathbf{S}_k\right)^\dagger \mathbf{S}_k^\top g(x_k) + \text{Null}(\mathbf{S}_k^\top \mathbf{H}(x_k)\mathbf{S}_k).
$$

Plugging the above into (44) gives

$$\mathbf{H}(x_k)(x - x_k) = -\frac{1}{\hat{L}}\mathbf{H}(x_k)\mathbf{S}_k\left(\mathbf{S}_k^\top \mathbf{H}(x_k)\mathbf{S}_k\right)^\dagger \mathbf{S}_k^\top g(x_k) + \mathbf{H}(x_k)\mathbf{S}_k\,\mathrm{Null}(\mathbf{S}_k^\top \mathbf{H}(x_k)\mathbf{S}_k)$$

$$\overset{(6)}{=} -\frac{1}{\hat{L}}\mathbf{H}(x_k)\mathbf{S}_k\left(\mathbf{S}_k^\top \mathbf{H}(x_k)\mathbf{S}_k\right)^\dagger \mathbf{S}_k^\top g(x_k). \tag{46}$$

Thus (8) is a solution to the above. If $\mathrm{Range}\,(\mathbf{S}_k) \subset \mathrm{Range}\,(\mathbf{H}_k(x_k))$ then $\mathbf{H}_k^\dagger(x_k)\mathbf{H}_k(x_k)\mathbf{S}_k = \mathbf{S}_k$ and the least norm solution is given by (8).

$\square$

## E  Proof of Theorem 2

*Proof.* Consider the iterates $x_k$ given by Algorithm 1 and let $\mathbb{E}_k\left[\cdot\right]$ denote the expectation conditioned on $x_k$, that is $\mathbb{E}_k\left[\cdot\right] = \mathbb{E}\left[\cdot \mid x_k\right]$. Setting $y = x_k$ in (4) and minimizing both sides[7] using (33) in Lemma 9, we obtain the inequality

$$f_* \geq f(x_k) - \frac{1}{2\hat{\mu}}\|g(x_k)\|^2_{\mathbf{H}^\dagger(x_k)}. \tag{47}$$

From (11) and (3) we have that

$$f(x_{k+1}) \leq f(x_k) - \frac{1}{2\hat{L}}\|g(x_k)\|^2_{\mathbf{S}_k(\mathbf{S}_k^\top \mathbf{H}(x_k)\mathbf{S}_k)^\dagger \mathbf{S}_k}. \tag{48}$$

Taking expectation conditioned on $x_k$ gives

$$\mathbb{E}_k\left[f(x_{k+1})\right] \leq f(x_k) - \frac{1}{2\hat{L}}\|g(x_k)\|^2_{\mathbf{G}(x_k)}. \tag{49}$$

Assumption 2 together with $\mathrm{Range}\,(\mathbf{H}(x_k)) = \mathrm{Range}\,\left(\mathbf{H}^{1/2}(x_k)\right)$ gives that

$$\mathbf{H}^{\dagger/2}(x_k)\mathbf{H}^{1/2}(x_k)g(x_k) = g(x_k), \tag{50}$$

where $\mathbf{H}^{\dagger/2}(x_k) = (\mathbf{H}^\dagger(x_k))^{1/2}$. Consequently

$$\|g(x_k)\|^2_{\mathbf{G}(x_k)} = \|g(x_k)\|^2_{\mathbf{H}^{\dagger/2}(x_k)\mathbf{H}^{1/2}(x_k)\mathbf{G}(x_k)\mathbf{H}^{1/2}(x_k)\mathbf{H}^{\dagger/2}(x_k)} \geq \rho(x_k)\|g(x_k)\|^2_{\mathbf{H}^\dagger(x_k)}, \tag{51}$$

where we used the definition (14) of $\rho(x_k)$ together with $\mathbf{H}^{\dagger/2}(x_k)g(x_k) \in \mathrm{Range}\,(\mathbf{H}(x_k))$ in the inequality. Using (51) and (47) in (49) gives

$$\mathbb{E}_k\left[f(x_{k+1})\right] \leq f(x_k) - \frac{\rho(x_k)}{2\hat{L}}\|g(x_k)\|^2_{\mathbf{H}^\dagger(x_k)} \tag{52}$$

$$\leq f(x_k) - \frac{\rho(x_k)\hat{\mu}}{\hat{L}}(f(x_k) - f_*). \tag{53}$$

Subtracting $f_*$ from both sides gives

$$\mathbb{E}_k\left[f(x_{k+1}) - f_*\right] \leq \left(1 - \rho(x_k)\frac{\hat{\mu}}{\hat{L}}\right)(f(x_k) - f_*). \tag{54}$$

Finally, since $x_k \in \mathcal{Q}$ from Lemma 3, we have that $\rho \leq \rho(x_k)$ and taking total expectation gives the result (15). $\square$

## F  Proof of Theorem 3

*Proof.* From (52) it follows that

$$\mathbb{E}\left[\|g(x_k)\|^2_{\mathbf{H}^\dagger(x_k)}\right] \overset{(52)}{\leq} \mathbb{E}\left[\frac{2\hat{L}}{\rho(x_k)}\left(f(x_k) - \mathbb{E}_k\left[f(x_{k+1})\right]\right)\right]$$

$$= \frac{2\hat{L}}{\rho(x_k)}\mathbb{E}\left[f(x_k) - f(x_{k+1})\right]$$

$$\overset{(14)}{\leq} \frac{2\hat{L}}{\rho}\mathbb{E}\left[f(x_k) - f(x_{k+1})\right]. \tag{55}$$

From (48) we have that

$$f(x_{k+1}) \leq f(x_k), \tag{56}$$

and thus

$$x_k \in \mathcal{Q} \quad \text{for all } k = 0, 1, 2, \ldots \tag{57}$$

Using the convexity of $f(x)$, for every $x_* \in \mathcal{X}_* := \arg\min f$ we get

$$
\begin{aligned}
f_* &\geq f(x_k) + \langle g(x_k), x_* - x_k \rangle \\
&\stackrel{(50)}{=} f(x_k) + \left\langle \mathbf{H}^{1/2}(x_k)\mathbf{H}^{\dagger/2}(x_k)g(x_k), x_* - x_k \right\rangle \\
&\geq f(x_k) - \|g(x_k)\|_{\mathbf{H}^{\dagger}(x_k)} \|x_k - x_*\|_{\mathbf{H}(x_k)} \\
&\stackrel{(57)}{\geq} f(x_k) - \|g(x_k)\|_{\mathbf{H}^{\dagger}(x_k)} \sup_{x \in \mathcal{Q}} \|x - x_*\|_{\mathbf{H}(x)},
\end{aligned}
$$

hence

$$f(x_k) - f_* \leq \|g(x_k)\|_{\mathbf{H}^{\dagger}(x_k)} \sup_{x \in \mathcal{Q}} \|x - x_*\|_{\mathbf{H}(x)}.$$

Taking infimum among all $x^* \in \mathcal{X}_*$ and using (17) we get

$$f(x_k) - f_* \leq \mathcal{R} \|g(x_k)\|_{\mathbf{H}^{\dagger}(x_k)}. \tag{58}$$

Hence by Jensen's inequality

$$
\begin{aligned}
(\mathbb{E}\left[f(x_k)\right] - f_*)^2 &\leq \mathbb{E}\left[(f(x_k) - f_*)^2\right] \\
&\stackrel{(58)}{\leq} \mathbb{E}\left[\mathcal{R}^2 \|g(x_k)\|^2_{\mathbf{H}^{\dagger}(x_k)}\right] \\
&\stackrel{(55)}{\leq} \frac{2\hat{L}\mathcal{R}^2}{\rho} \mathbb{E}\left[f(x_k) - f(x_{k+1})\right].
\end{aligned}
\tag{59}
$$

Now we put everything together:

$$
\begin{aligned}
\frac{1}{\mathbb{E}\left[f(x_{k+1}) - f_*\right]} - \frac{1}{\mathbb{E}\left[f(x_k) - f_*\right]} &= \frac{\mathbb{E}\left[f(x_k) - f(x_{k+1})\right]}{\mathbb{E}\left[f(x_{k+1}) - f_*\right]\mathbb{E}\left[f(x_k) - f_*\right]} \\
&\stackrel{(56)}{\geq} \frac{\mathbb{E}\left[f(x_k) - f(x_{k+1})\right]}{(\mathbb{E}\left[f(x_k) - f_*\right])^2} \\
&\stackrel{(59)}{\geq} \frac{\rho}{2\hat{L}\mathcal{R}^2}.
\end{aligned}
\tag{60}
$$

Summing up (60) for $k = 0, \ldots, T-1$ and using telescopic cancellation we get

$$\frac{\rho T}{2\hat{L}\mathcal{R}^2} \leq \frac{1}{\mathbb{E}\left[f(x_T) - f_*\right]} - \frac{1}{\mathbb{E}\left[f(x_0) - f_*\right]} \leq \frac{1}{\mathbb{E}\left[f(x_T) - f_*\right]}, \tag{61}$$

which after re-arranging concludes the proof. $\qquad \square$

## G  Proof of Lemma 6

*Proof.* If (19) holds then by taking orthogonal complements we have that

$$\text{Range}\left(\mathbf{H}(x_k)\right) = \text{Null}\left(\mathbf{H}(x_k)\right)^{\perp} = \text{Null}\left(\mathrm{E}[\hat{\mathbf{P}}(x_k)]\right)^{\perp}, \tag{62}$$

and consequently

$$
\begin{aligned}
\rho(x_k) &\stackrel{(14)+(62)}{=} \min_{v \in \text{Null}(\mathrm{E}[\hat{\mathbf{P}}(x_k)])^{\perp}} \frac{\left\langle \mathbf{H}^{1/2}(x_k)\mathbf{G}(x_k)\mathbf{H}^{1/2}(x_k)v, v \right\rangle}{\|v\|_2^2} \\
&= \min_{v \in \text{Null}(\mathrm{E}[\hat{\mathbf{P}}(x_k)])^{\perp}} \frac{\left\langle \mathrm{E}[\hat{\mathbf{P}}(x_k)]v, v \right\rangle}{\|v\|_2^2} = \lambda^+_{\min}(\mathrm{E}[\hat{\mathbf{P}}(x_k)]) > 0.
\end{aligned}
$$

$$\square$$

# H Proof of Lemma 7

*Proof.* Let $\mathcal{X}_{\mathbf{S}}$ be a random subset of $\mathbb{R}^d$, where $\mathbf{S} \sim \mathcal{D}$. We define stochastic intersection of $\mathcal{X}_{\mathbf{S}}$:

$$\bigcap_{\mathbf{S} \sim \mathcal{D}} \mathcal{X}_{\mathbf{S}} = \left\{ x \in \mathbb{R}^d : x \in \mathcal{X}_{\mathbf{S}} \text{ with probability } 1 \right\}. \tag{63}$$

Using this definition for $\text{Null}(\mathbf{G}_k)$ we have

$$\begin{aligned}
\text{Null}\left(\mathbf{G}_k\right) &= \text{Null}\left( \mathbb{E}_{\mathbf{S} \sim \mathcal{D}} \left[ \mathbf{S} \left( \mathbf{S}^\top \mathbf{H}(x_k) \mathbf{S} \right)^\dagger \mathbf{S}^\top \right] \right) \\
&= \bigcap_{\mathbf{S} \sim \mathcal{D}} \text{Null}\left( \mathbf{S} \left( \mathbf{S}^\top \mathbf{H}(x_k) \mathbf{S} \right)^\dagger \mathbf{S}^\top \right),
\end{aligned} \tag{64}$$

where the last equality follows from the fact that $\mathbf{S} \left( \mathbf{S}^\top \mathbf{H}(x_k) \mathbf{S} \right)^\dagger \mathbf{S}^\top$ is a symmetric positive semidefinite matrix. From the properties of pseudoinverse it follows that

$$\text{Null}\left( \left( \mathbf{S}^\top \mathbf{H}(x_k) \mathbf{S} \right)^\dagger \right) = \text{Null}\left( \mathbf{S}^\top \mathbf{H}(x_k) \mathbf{S} \right) = \text{Null}\left( \mathbf{S} \right),$$

thus, we can apply Lemma 10 and obtain

$$\text{Null}\left( \mathbf{S} \left( \mathbf{S}^\top \mathbf{H}(x_k) \mathbf{S} \right)^\dagger \mathbf{S}^\top \right) = \text{Null}\left( \mathbf{S}^\top \right). \tag{65}$$

Furthermore,

$$\begin{aligned}
\text{Null}\left(\mathbf{G}_k\right) &\overset{(64)}{=} \bigcap_{\mathbf{S} \sim \mathcal{D}} \text{Null}\left( \mathbf{S} \left( \mathbf{S}^\top \mathbf{H}(x_k) \mathbf{S} \right)^\dagger \mathbf{S}^\top \right) \\
&\overset{(65)}{=} \bigcap_{\mathbf{S} \sim \mathcal{D}} \text{Null}\left( \mathbf{S}^\top \right) \\
&= \bigcap_{\mathbf{S} \sim \mathcal{D}} \text{Null}\left( \mathbf{S} \mathbf{S}^\top \right) \\
&= \text{Null}\left( \mathbb{E}_{\mathbf{S} \sim \mathcal{D}} \left[ \mathbf{S} \mathbf{S}^\top \right] \right).
\end{aligned} \tag{66}$$

From (20) and (66) it follows that

$$\text{Null}\left(\mathbf{G}_k\right) \subset \text{Null}\left(\mathbf{H}(x_k)\right) = \text{Null}\left( \mathbf{H}^{1/2}(x_k) \right), \tag{67}$$

hence, Lemma 10 implies that

$$\text{Range}\left(\mathbf{H}(x_k)\right) = \text{Range}\left( \mathbf{H}^{1/2}(x_k) \mathbf{G}_k \mathbf{H}^{1/2}(x_k) \right), \tag{68}$$

which concludes the proof.

$\square$

# I Proof of Lemma 8

*Proof.* Using Taylor's theorem, for every $x, y \in \mathcal{Q}$ we have that

$$f(x) = f(y) + \langle g(y), x - y \rangle + \int_{t=0}^{1} (1-t) \|x - y\|_{\mathbf{H}(y + t(x-y))}^2 dt. \tag{69}$$

Comparing the above with (3) we have that

$$\frac{\hat{L}}{2} \|x - y\|_{\mathbf{H}(y)}^2 \geq \int_{t=0}^{1} (1-t) \|x - y\|_{\mathbf{H}(y + t(x-y))}^2 dt, \quad \forall x, y \in \mathcal{Q}, \ x \neq y. \tag{70}$$

Let $x \neq y$. Since we assume that $\|x - y\|_{\mathbf{H}(y)}^2 \neq 0$ we have that the relative smoothness constant satisfies

$$\frac{\hat{L}}{2} = \max_{x,y \in \mathcal{Q}} \int_{t=0}^{1} \frac{(1-t) \|x - y\|_{\mathbf{H}(y + t(x-y))}^2}{\|x - y\|_{\mathbf{H}(y)}^2} dt. \tag{71}$$

Let $z_t = y + t(x - y)$. Substituting $x - y = (z_t - y)/t$ in the above gives the equality in (21). Following an analogous argument for the relative convexity constant $\hat{\mu}$ gives the equality in (21).

Since $f(x)$ is convex, the set $\mathcal{Q}$ is convex and thus $z_t \in \mathcal{Q}$ for all $t \in [0, 1]$. By alternating the order of the maximization and integral in (21) that

$$\frac{\hat{L}}{2} \overset{(21)}{\leq} \int_{t=0}^{1} (1-t) \max_{x,y \in \mathcal{Q}} \frac{\|z_t - y\|_{\mathbf{H}(z_t)}^2}{\|z_t - y\|_{\mathbf{H}(y)}^2} dt$$

$$\overset{z_t \in \mathcal{Q}}{\leq} \int_{t=0}^{1} (1-t) dt \max_{x,y \in \mathcal{Q}} \frac{\|x - y\|_{\mathbf{H}(x)}^2}{\|x - y\|_{\mathbf{H}(y)}^2} = \frac{1}{2} \max_{x,y \in \mathcal{Q}} \frac{\|x - y\|_{\mathbf{H}(x)}^2}{\|x - y\|_{\mathbf{H}(y)}^2}.$$

Following an analogous argument for the relative convexity constant $\hat{\mu}$ we have that

$$\frac{\hat{\mu}}{2} \overset{(22)}{\geq} \int_{t=0}^{1} (1-t) \min_{x,y \in \mathcal{Q}} \frac{\|z_t - y\|_{\mathbf{H}(z_t)}^2}{\|z_t - y\|_{\mathbf{H}(y)}^2} dt$$

$$\overset{z_t \in \mathcal{Q}}{\geq} \int_{t=0}^{1} (1-t) dt \min_{x,y \in \mathcal{Q}} \frac{\|x - y\|_{\mathbf{H}(x)}^2}{\|x - y\|_{\mathbf{H}(y)}^2} = \frac{1}{2} \frac{1}{\max_{x,y \in \mathcal{Q}} \frac{\|x-y\|_{\mathbf{H}(x)}^2}{\|x-y\|_{\mathbf{H}(y)}^2}}.$$

$\square$

## J  Proof of Corollary 1

*Proof.* Using that

$$0 < d_i^\top \mathbf{H}(x) d_i \leq d_i \mathbf{U} d_i, \tag{72}$$

which follows from $\mathbf{H} \preceq \mathbf{U}$ and our assumption that $d_i^\top \mathbf{H}(x) d_i \neq 0$, we have that

$$\mathbf{G}(x) = \mathbb{E}_k \left[ \mathbf{S}(\mathbf{S}^\top \mathbf{H}(x)\mathbf{S})^\dagger \mathbf{S}^\top \right] = \sum_{i=1}^{d} \frac{d_i \mathbf{U} d_i}{\text{Trace}\left(\mathbf{D}^\top \mathbf{U} \mathbf{D}\right)} \frac{d_i d_i^\top}{d_i^\top \mathbf{H}(x) d_i}$$

$$\overset{(72)}{\succeq} \frac{1}{\text{Trace}\left(\mathbf{D}^\top \mathbf{U} \mathbf{D}\right)} \sum_{i=1}^{d} d_i d_i^\top = \frac{1}{\text{Trace}\left(\mathbf{D}^\top \mathbf{U} \mathbf{D}\right)} \mathbf{D} \mathbf{D}^\top. \tag{73}$$

Furthermore since $\mathbf{D}$ is invertible we have by Lemma 10 that

$$\text{Range}\left(\mathbf{H}^{1/2}(x) \mathbf{D} \mathbf{D}^\top \mathbf{H}^{1/2}\right) = \text{Range}\left(\mathbf{H}^{1/2}(x)\right) = \text{Range}\left(\mathbf{H}(x)\right). \tag{74}$$

And thus from Lemma 6 we have that

$$\rho = \min_{x \in \mathcal{Q}} \lambda_{\min}^+(\hat{\mathbf{P}}(x)) \overset{(18)}{\geq} \min_{x \in \mathcal{Q}} \frac{\lambda_{\min}^+(\mathbf{H}^{1/2}(x)\mathbf{D}\mathbf{D}^\top \mathbf{H}^{1/2}(x))}{\text{Trace}\left(\mathbf{D}^\top \mathbf{U} \mathbf{D}\right)}. \tag{75}$$

$\square$

## K  Proof of Proposition 1

*Proof.* The gradient and Hessian of (26) are given by

$$g(x) = \frac{1}{n} \sum_{i=1}^{n} a_i \phi_i'(a_i^\top x) + \lambda x = \frac{1}{n} \mathbf{A} \Phi'(\mathbf{A}^\top x) + \lambda x, \tag{76}$$

$$\mathbf{H}(x) = \frac{1}{n} \sum_{i=1}^{n} a_i a_i^\top \phi_i''(a_i^\top x) + \lambda \mathbf{I} = \frac{1}{n} \mathbf{A} \Phi''(\mathbf{A}^\top x) \mathbf{A}^\top + \lambda \mathbf{I}, \tag{77}$$

where

$$\Phi'(\mathbf{A}^\top x) := [\phi_i'(a_1^\top x), \ldots, \phi_i'(a_n^\top x)] \in \mathbb{R}^n, \tag{78}$$

$$\Phi''(\mathbf{A}^\top x) := \text{diag}\left(\phi_i''(a_1^\top x), \ldots, \phi_i''(a_n^\top x)\right). \tag{79}$$

Consequently the $g(x) \in \text{Range}\left(\mathbf{H}(x)\right)$ for all $x \in \mathbb{R}^d$.

Using Lemma 8 and (77) we have that

$$
\begin{aligned}
\hat{L} \quad &\leq \quad \max_{y,z \in \mathbb{R}^d} \frac{\|y - z\|^2_{\frac{1}{n}\mathbf{A}\Phi''(\mathbf{A}^\top y)\mathbf{A}^\top + \lambda\mathbf{I}}}{\|y - z\|^2_{\frac{1}{n}\mathbf{A}\Phi''(\mathbf{A}^\top z)\mathbf{A}^\top + \lambda\mathbf{I}}} \\
&\overset{(25)}{\leq} \quad \max_{y,z \in \mathbb{R}^d} \frac{\|y - z\|^2_{\frac{\ell}{n}\mathbf{A}\mathbf{A}^\top + \lambda\mathbf{I}}}{\|y - z\|^2_{\frac{u}{n}\mathbf{A}\mathbf{A}^\top + \lambda\mathbf{I}}} \\
&= \quad \max_{y,z \in \mathbb{R}^d} \frac{\|y - z\|^2_{\frac{\ell - u}{n}\mathbf{A}\mathbf{A}^\top} + \|y - z\|^2_{\frac{u}{n}\mathbf{A}\mathbf{A}^\top + \lambda\mathbf{I}}}{\|y - z\|^2_{\frac{u}{n}\mathbf{A}\mathbf{A}^\top + \lambda\mathbf{I}}} \\
&= \quad 1 + \max_{y,z \in \mathbb{R}^d} \frac{\|y - z\|^2_{\frac{\ell - u}{n}\mathbf{A}\mathbf{A}^\top}}{\|y - z\|^2_{\frac{u}{n}\mathbf{A}\mathbf{A}^\top + \lambda\mathbf{I}}}
\end{aligned}
\tag{80}
$$

Now note that

$$
\begin{aligned}
\max_{y,z \in \mathbb{R}^d} \frac{\|y - z\|^2_{\frac{\ell - u}{n}\mathbf{A}\mathbf{A}^\top}}{\|y - z\|^2_{\frac{u}{n}\mathbf{A}\mathbf{A}^\top + \lambda\mathbf{I}}} \quad &= \quad \frac{1}{\displaystyle\min_{y,z \in \mathbb{R}^d} \frac{\|y - z\|^2_{\frac{u}{n}\mathbf{A}\mathbf{A}^\top + \lambda\mathbf{I}}}{\|y - z\|^2_{\frac{\ell - u}{n}\mathbf{A}\mathbf{A}^\top}}} \\
&= \quad \frac{1}{\dfrac{u}{\ell - u} + \lambda \displaystyle\min_{y,z \in \mathbb{R}^d} \frac{\|y - z\|^2_2}{\|y - z\|^2_{\frac{\ell - u}{n}\mathbf{A}\mathbf{A}^\top}}} \\
&= \quad \frac{1}{\dfrac{u}{\ell - u} + \dfrac{n\lambda}{\ell - u}\dfrac{1}{\sigma^2_{\max}(\mathbf{A})}},
\end{aligned}
\tag{81}
$$

where we used that

$$
\min_{y,z \in \mathbb{R}^d} \frac{\|y - z\|^2_2}{\|y - z\|^2_{\mathbf{A}\mathbf{A}^\top}} = \frac{1}{\max_{y,z \in \mathbb{R}^d} \frac{\|y-z\|^2_{\mathbf{A}\mathbf{A}^\top}}{\|y-z\|^2_2}} = \frac{1}{\sigma^2_{\max}(\mathbf{A})}.
\tag{82}
$$

Inserting (81) into (80) gives

$$
\hat{L} \leq 1 + \frac{\ell - u}{u + \frac{n\lambda}{\sigma^2_{\max}(\mathbf{A})}} = \frac{\ell\sigma^2_{\max}(\mathbf{A}) + n\lambda}{u\sigma^2_{\max}(\mathbf{A}) + n\lambda}.
$$

The bounds for $\hat{\mu}$ follows from (22).

Finally turing to Lemma 7 we have that (6) holds since $\mathbf{H}(x_k)$ is positive definite and by Lemma 10, and (20) holds by our assumption that $\mathbb{E}\left[\mathbf{S}\mathbf{S}^\top\right]$ is invertible. Thus by Lemma 7 we have that $\rho > 0$ and the total complexity result in Theorem 2 holds. $\qquad\square$

## L  Uniform single coordinate sketch

Further to our results on using single column sketches with non-uniform sampling in Corollary 1, here we present the case for uniform sampling that does not rely on the Hessian having a uniform upper bound as is assumed in Corollary 1. Let $\mathbf{H}_{ii}(x) \coloneqq e_i^\top \mathbf{H}(x)e_i$ and $g_i(x) \coloneqq e_i^\top g(x)$. In this case (8) is given by

$$
x_{k+1} = x_k - \frac{g_i(x_k)}{\hat{L}\mathbf{H}_{ii}(x_k)}e_i.
\tag{83}
$$

**Algorithm 2** RSNxls: Randomized Subspace Newton with exact Line-Search

1: **input:** $x_0 \in \mathbb{R}^d$
2: **parameters:** $\mathcal{D}$ = distribution over random matrices
3: **for** $k = 0, 1, 2, \ldots$ **do**
4:     $\mathbf{S}_k \sim \mathcal{D}$
5:     $\lambda_k = -\left(\mathbf{S}_k^\top \mathbf{H}(x_k)\mathbf{S}_k\right)^\dagger \mathbf{S}_k^\top g(x_k)$
6:     $d_k = \mathbf{S}_k \lambda_k$
7:     $t_k = \operatorname{argmin}_{t \in \mathbb{R}} f(x_k + td_k)$
8:     $x_{k+1} = x_k + t_k d_k$
9: **output:** last iterate $x_k$

**Corollary 2.** *Let* $\mathbb{P}[\mathbf{S}_k = e_i] = \frac{1}{d}$ *and let*

$$\alpha = \min_{x \in \mathbb{R}^d} \min_{w \in \operatorname{Range}(\mathbf{H}(x))} \frac{\|w\|_{\mathbf{Diag}(\mathbf{H}(x))^{-1}}^2}{\|w\|_{\mathbf{H}^\dagger(x)}^2}.$$

*Under the assumptions of Theorem 2 we have that Algorithm 1 converges according to*

$$\mathbb{E}\left[f(x_k) - f_*\right] \leq \left(1 - \frac{\alpha}{d}\frac{\hat{\mu}}{\hat{L}}\right)^k (f(x_0) - f_*).$$

*Proof.* It follows by direct computation that

$$\mathbf{G}(x) = \mathbb{E}_k\left[\mathbf{S}(\mathbf{S}^\top \mathbf{H}(x)\mathbf{S})^\dagger \mathbf{S}^\top\right] = \frac{1}{d}\sum_{i=1}^{d} \frac{e_i e_i^\top}{\mathbf{H}_{ii}(x)} = \frac{1}{d}\mathbf{Diag}\left(\mathbf{H}(x)\right)^{-1}.$$

Thus from the definition (14) we have

$$\rho = \frac{1}{d}\min_{x \in \mathbb{R}^d} \min_{v \in \operatorname{Range}(\mathbf{H}(x))} \frac{\left\langle \mathbf{H}^{1/2}(x)\mathbf{Diag}\left(\mathbf{H}(x)\right)^{-1}\mathbf{H}^{1/2}(x)v, v\right\rangle}{\|v\|_2^2}.$$

Since $\operatorname{Range}\left(\mathbf{H}^{\dagger/2}(x)\right) = \operatorname{Range}\left(\mathbf{H}(x)\right)$ and $v \in \operatorname{Range}\left(\mathbf{H}(x)\right)$ we can re-write $v = \mathbf{H}^{\dagger/2}(x)w$ where $w \in \operatorname{Range}\left(\mathbf{H}(x)\right)$ and consequently

$$\rho = \frac{1}{d}\min_{x \in \mathbb{R}^d} \min_{w \in \operatorname{Range}(\mathbf{H}(x))} \frac{\left\langle \mathbf{Diag}\left(\mathbf{H}(x)\right)^{-1}\mathbf{H}^{1/2}(x)\mathbf{H}^{\dagger/2}(x)w, \mathbf{H}^{1/2}(x)\mathbf{H}^{\dagger/2}(x)w\right\rangle}{\|w\|_{\mathbf{H}^\dagger(x)}^2}$$

$$\overset{\mathbf{H}^{1/2}(x)\mathbf{H}^{\dagger/2}(x)w=w}{=} \frac{1}{d}\min_{x \in \mathbb{R}^d} \min_{w \in \operatorname{Range}(\mathbf{H}(x))} \frac{\left\langle \mathbf{Diag}\left(\mathbf{H}(x)\right)^{-1}w, w\right\rangle}{\langle \mathbf{H}(x)w, w\rangle_2^2} := \frac{\alpha}{d}.$$

$\square$

# M    Experimental details

All tests were performed in MATLAB 2018b on a PC with an Intel quad-core i7-4770 CPU and 32 Gigabyte of DDR3 RAM running Ubuntu 18.04.

## M.1    Sketched Line-Search

In order to speed up convergence we can modify Algorithm 1 by introducing an exact Line-Search and obtain Algorithm 2.

In this section we focus on heuristics for performing an exact Line-Search under the assumption that our direction is of the form $d = \mathbf{S}\lambda$. This allows us to only work with sketched gradients

---

**Algorithm 3** Generic Line Search - Pseudocode

---

1: **input:** increasing continuous function $l : \mathbb{R} \to \mathbb{R}$ with $l(0) < 0$ and at least one root $t^* \in \mathbb{R}_+$
2: **tolerance:** $\epsilon > 0$
3: set $[a, b] \leftarrow [0, 1]$
4: **while** $l(b) < -\epsilon$
5:      choose $t > b$             $\triangleright$ either fixed enlargement ($t = 2b$) or via spline extrapolation
6:      set $[a, b] \leftarrow [b, t]$
7: **endwhile**          $\triangleright$ end of first phase: either $|l(b)| \leq \epsilon$ or $l(a) < 0 < \epsilon \leq l(b)$, i.e. $t^* \in [a, b]$
8: set $t \leftarrow b$
9: **while** $|l(t)| > \epsilon$
10:      **if** $l(t) < 0$
11:          $[a, b] \leftarrow [t, b]$
12:      **else** $l(t) > 0$
13:          $[a, b] \leftarrow [a, t]$
14:      **endif**
15: choose $t$ with $a < t < b$      $\triangleright$ either middle of interval ($t = \frac{a+b}{2}$) or via spline interpolation
16: **endwhile**                                  $\triangleright$ end of second phase
17: **output:** $t > 0$ with $|l(t)| \leq \epsilon$

---

and sketched Hessians. This potentially allows for significant computational savings. Specifically consider the problem of finding

$$t^* := \operatorname{argmin}_{t \in \mathbb{R}} f(x + td), \tag{84}$$

which is, for differentiable and convex $f$, equivalent to finding a root of the objectives first derivative. Defining

$$l(t) := \frac{\partial f(x + td)}{\partial t} = d^\top g(x + td) = \lambda^\top (\mathbf{S}^\top g(x + td)) \tag{85}$$

gives us the task of solving

$$l(t^*) = 0 \tag{86}$$

and differentiating once more

$$l'(t) = \frac{\partial^2 f(x + td)}{\partial^2 t} = d^\top \mathbf{H}(x + td)d = \lambda^\top (\mathbf{S}^\top \mathbf{H}(x + td)\mathbf{S})\lambda, \tag{87}$$

reveals that we do not need full, but only sketched gradient and Hessian access, in order to evaluate $l$ respectively $l'$. Note that the evaluation of

$$l(0) = \lambda^\top \mathbf{S}^\top g(x)$$
$$l'(0) = \lambda^\top (\mathbf{S}^\top \mathbf{H}(x)\mathbf{S})\lambda \tag{88}$$

are essentially a by-product from the computation of $\lambda$ in Algorithm 2 and therefore add almost no computational cost. Furthermore, if $f$ is convex and $\lambda = -(\mathbf{S}^\top \mathbf{H}(x)\mathbf{S})^\dagger \mathbf{S}^\top g(x)$ is given , then

$$l(0) = -g(x)^\top \mathbf{S}(\mathbf{S}^\top \mathbf{H}(x)\mathbf{S})^\dagger \mathbf{S}^\top g(x) \leq 0 \tag{89}$$

implies that $d$ is a weak descent direction of $f$. Since in this case, $l(0) = 0$ implies $t^* = 0$, let us focus on the situation that we actually have a strong descent direction, i.e. that

$$l(0) < 0 \tag{90}$$

is satisfied. The line-search 3 ensures an output $t > 0$ satisfying $|l(t)| \leq \epsilon$ and is best explained by strengthening Step 4 of (3) to "**while** $l(b) < 0$", as this would ensure that the final values of $a$ and $b$ box the minimum $t^* \in [a, b]$: The first phase is to identify an interval $[a, b]$ with $0 \leq a < b$ such that

$$l(a) < 0 \leq l(b) \tag{91}$$

which guarantees the existence of at least one minimum $t^* \in [a, b]$. In the second phase, we can then decrease the intervals length with $a \leq \bar{a} < \bar{b} \leq b$ such that $0 \leq l(t) \leq \epsilon$ is satisfied for all $t \in [\bar{a}, \bar{b}]$ and some given tolerance $\epsilon > 0$. Both steps should be safeguarded and can be assisted by using cubic splines inter- or extrapolating $l(t)$. This approach has the potential of reducing computational costs and the benefit of avoiding function evaluations of $f$ entirely.

## Footnotes

[7]Note that $x^* \in \mathcal{Q}$ but the global minimizer of (33) is not necessarily in $\mathcal{Q}$. This is not an issue, since the global minima is a lower bound on the minima constrained to $\mathcal{Q}$.