[Reviews · NeurIPS 2019]

Reviewer 1



The paper introduces a new family of randomized Newton methods, based on a prototypical Hessian sketching scheme to reduce the memory and arithmetic costs. Clearly, the idea of using a randomized sketch for the Hessian is not new. However, the paper extends the known results in a variety of ways: The proposed method gets linear convergence rate 1) under the relative smoothness and the relative convexity assumptions (and the method is still scale-invariant). 2) The method works practically with all sketching techniques and 3) for any sketch size. These results also include the known results for the Newton method as a special case. The related work is adequately cited, the similar approaches from the existing literature and their weaknesses are discussed in a short but concise discussion in the paper. The results are technically sound, and these results are explained in a simple and clear way. The proposed method outperforms the state of the art. The experiments section is well-presented, and the results there support the theoretical findings. Empirical evidence is limited to a logistic regression setup (with a variety of real data). It could have been expanded for some other settings, either in the main text or supplements. Overall, I enjoyed reading this submission. The paper is very clear about its focus and contributions, and all technical results and the figures are accompanied by discussions and take-home messages. The proposed method can be attractive to the practitioners since it is scale invariant (hence no tuning or preconditioning) and it outperforms the baseline. I recommend the acceptance of this paper. Some minor comments: - There is a typo in footnote 4 on page 5. - "n" is defined as the Newton direction in (5). It is also used to denote the number of data points in pages 6-7. Consider changing one of these to avoid confusion. - line 259: "out run" or "outrun"? - [12] and [13] are the same papers the bibliography ========== After author feedback ========== I have read the author feedback and takin it into account in my final scores.

Reviewer 2



The main idea of this paper is to develop a sketching Newton algorithm for a class of strongly convex and Lipschitz gradient functions in local norms, which has been studied in [20], but for proximal-Newton-type methods. The proposed algorithm converges to the solution of the problem at a global linear or sublinear rate. Such a rate depends on the choice of sketching matrix. The following points are my concerns: -- The efficiency of the method depends on the choice of sketching matrix S. While Theorem 2 shows a linear convergence rate, it depends on \rho(x), a strong condition, and the authors did not show when it is strictly positive. It is also interesting to see a theoretical trade-off between the dimension of the sketching matrix and the contraction factor in Theorem 2 (at least on a particular class of functions). -- Clearly, the linear convergence stated in Theorem 2 is not surprised under Assumptions 1 and 2, which is similar to gradient methods. This has been stated in [20] as an inexact/quasi Newton method, though not thoughtfully discussed. This certainly makes the contribution of the paper to be minor. The authors should carefully discuss this point. --- The main technical proofs of the paper appear to be correct. I have checked the main proofs in the supplementary. -- The paper is generally well-written and clearly structured. There are some small possible mistakes. Firstly, in equation (52) all norm should be replaced by square of norm. Secondly, the second equation of (76) doesn’t look right to me because Pˆ(x) = H1/2G(x)H1/2 by definition. -- The paper provides only one class of experiment on a regularized logistic problem. This problem satisfies the assumptions studied in other subsampled and sketching methods such as [6,28]. Why the authors do not compare the proposed method with these existing ones? -- The proof of Lemma 3 can be simplified as: f(xk+1) = minλ T(xk +Skλ;xk) ≤ T(xk;xk) = f(xk). -- The result of Theorem 2 is not perfect. There is no theory to gurantee that rho > 0. In addition, it is not clear why rho <= 1. The author should also indicate how the sketch size influences rho. --The assumption (17) of Theorem 3 seems to be too strong. A clear explanation is needed.

Reviewer 3



The biggest hole I can find here is a detailed, clear comparison with Pilanci/Wainwright 2015 work on subsampled / sketched Newton, where they give convergence rates for both the sketch size > rank and sketch size < rank regime. As far as I can see, this work is not particularly novel when compared with the Pilanci/Wainwright work, although I can believe that it rose independently (it does not seem exactly the same). Therefore a clear discussion of the comparison would help rate originality and significance. Quality: The convergence rate looks more similar to that of a first-order method than a second-order method, and I believe if the sketch size goes to 1, then the method should reduce to gradient descent. (Newton should be q-quadratic) Therefore the rates themselves are not super impressive. However, the math is clean and the few lemmas I checked, the proof looks sound. Clarity: I think the mathematical contributions are very clearly explained and intuitive, which is the paper's main strength. In particular, the interpretation of the sketched step as a randomized projection, followed by simple linear algebra consequences, is a very clean way of analyzing the convergence behavior. Comments after rebuttal: - Upon reading the rebuttal and rereading the main paper, I do think I misunderstood some key parts of their algorithm, and the comment they made that gradient descent may not even descend in this framework is true. Hopefully the authors can find a way to emphasize this more when proposing the method? (Maybe with an example with all dimensions labeled?) - The point about the square root Newton in P/W work makes sense, and it does seem that they are mostly focused on local convergence rates, whereas the authors here are focused on global convergence rates. With the promised included discussion, I think the novelty is clearer. - I do feel that the exact line search (which is not terribly computationally burdensome but is somewhat unconventional) should be discussed more in the main text rather than the appendix. I have increased the overall score.

[Author Response · NeurIPS 2019]

Dear reviewers, thank you for taking the time to review our paper. We have addressed your main questions in **A1**, **A2** and **A3**, and your remaining questions below. We especially thank **Rev1** for his/her thoughtful and encouraging remarks. All issues raised are easy to address. We will incorporate all of your suggestions.

**A1: Rank assumption on the Hessian.** Corollary 1 gives an example where the Hessian need not have full rank to satisfy our assumptions. Indeed whenever the sketching matrix $\mathbf{S}_k = s_k \in \mathbb{R}^d$ is a column vector, eq (9) and (20) hold trivially so long as $s_k^\top \mathbf{H}_k s_k \neq 0$. This can hold for rank deficient Hessian matrices for instance when the diagonal has no zero elements and when $s_k$ are random unit coordinate vectors.

**A2: Novelty related to Pilanci/Wainwright's work.** There are many key differences between RSN and Newton Sketch (NS) [20]. First, they are simply different algorithms. The sketching method underlying NS relies on having at hand the square root of the Hessian. In contrast, RSN uses a random subspace constraint to sketch the Hessian and thus needs no square root. Furthermore, NS requires full gradient and function evaluations, while RSN only needs a sketched gradient and requires no function evaluations. The convergence proof of NS requires a sketch size proportional to $\epsilon^{-2}$, an unknown universal constant and global spectral properties of the Hessian; see equations (12) and (19) in [30]. Thus this required sketch size could be as large as $d$ (or larger, which makes the results vacuous). This is because the theory of NS builds upon the theory of "one shot" sketching techniques. Furthermore, they do not establish linear convergence rates[1]. In contrast, we establish linear convergence which can hold in the rank deficient case and for every sketch size. We achieve this by entirely bypassing the theory of one shot sketches, showing it is not at all necessary. This in turn gives us the freedom of choosing the sketch size arbitrarily and allows us to apply RSN to large scale problems no matter how large the dimension. We will include this discussion in the paper.

**A3: Bounding** $0 < c \leq \rho \leq 1$. The bound $\rho \leq 1$ follows from Lemma 7 since $\rho(x_k) \leq 1$ for all $k$. We can guarantee that $\rho$ is bounded away from zero $\rho \geq c > 0$ if we use the common assumption that $f(x)$ is $L$–smooth and $m$–strongly convex. This follows under the conditions of Lemma 7 since:
$\rho(x) \geq m\lambda_{\min}^+ \left( \mathbb{E}\left[ \mathbf{S}(\mathbf{S}^\top \mathbf{H}(x)\mathbf{S})^\dagger \mathbf{S}^\top \right] \right) \geq \frac{m}{L}\beta$, where $\beta := \lambda_{\min}^+ \left( \mathbb{E}\left[ \mathbf{S}(\mathbf{S}^\top\mathbf{S})^\dagger\mathbf{S}^\top \right] \right)$. The right-hand side is a fixed positive constant independent on $x$, thus $\rho \geq \frac{m}{L}\beta > 0$. We can even relax the strongly convex assumption since only $\lambda_{\min}^+ (\mathbf{H}(x))$ needs to be uniformly bounded away from zero (the spectral gap must be lower bounded). Furthermore, $\beta$ is known for many distributions, e.g. for Gaussians and the family of randomized orthogonal sketches (Section A.1 in [12]) we have $\beta = \frac{s}{d}$, where $s$ is the sketch size and $d$ the dimension. Thus $\rho \geq \frac{m}{L}\frac{s}{d}$ and $\rho$ is at least linearly increasing in $s$. We will now include this lower bound.

**Rev2.** *Theorem 2 is not surprised ... This has been stated in [20] as an inexact Newton method.* Our RSN method is not an inexact Newton method since we do not need to guarantee that the quadratic upper bound is minimized to within a given accuracy threshold. In no way has RSN been stated/analysed in [20].

**Q1. Assumption (17) ... seems to be too strong.** For convex functions, this assumption is equivalent to $f$ being lower bounded, which is a trivial assumption, since otherwise $f$ is a linear function.

**Q2. Eq (52) and Eq (76).** Thank you, we have fixed the squared norms and (76) should be an inequality.

**Q3. Same assumptions as [6, 28]. Why not compare?** S-Newton in [6] is based on subsampling and has no dimension reduction: it is targeted at large $n$ and small $d$. The opposite setting of RSN. Also, subsampling and be applied in conjunction with our technique. The method in [28] is for solving constraint linear least square, not general optimization smooth and convex optimization.

**Rev3.** *"Newton should be q-quadratic. Therefore ... not super impressive."* There exists only semi-local quadratic convergence for Newton based methods. For global convergence, linear rates are as good as it gets.

**Q1.** *Comment about parameter tuning.* We apologize, but we did not understand your question/comment.

**Q2.** *Where is the proposed line search strategy.* It is in Algorithm 3 in the supp. material as stated on lines 242–243 of the main paper. Our line search does not require function evaluations, but only sketched gradients and sketched Hessian. Since the sketched Hessian is already available from the RSN update, our line search is computationally much cheaper than the standard Armijo method.

**Q3.** *Assumption 2 seems too strong.* Assumption 2 does not hold for $x_1^2 + \text{huber}_1(x_2)$ but neither is this a twice differentiable function, thus one cannot apply Newton type methods. Assumption 2 is necessary to guarantee that the Newton direction is well defined, see Lemma 9.

**Q4.** *The assumption does not hold for generalized linear models if ...* This assumption holds for all convex generalized linear models independently of the rank of $\mathbf{A}$ and the regularization parameter. This follows from examining the gradient and Hessian in (77) and (78) in the supp material and using standard linear algebra results such as Lemma 10. We will clarify this point and include the proof of this claim in the supp material.

**Q5.** *Theorem 2: Is this not the usual gradient descent rate?* Please see lines 219–225. In particular, Theorems 2 and 3 rely on relative smoothness and convex assumptions. Under these assumptions, it is not known if gradient descent converges.

**Q6.** *Is it possible that $\lambda_{\min}$ ... is larger with sketching than without?* Yes if $\mathbf{D}$ is a preconditioner $\mathbf{D} \approx \mathbf{U}^{-1}$.

## Footnotes

[1]See Theorem 2 in [10], where in the number of iterations $T$ is lower bounded by a constant term


[Meta-Review · NeurIPS 2019]

This paper considers a class of randomized subspace Newton methods, allowing sketching of the Hessian from arbitrary subspaces, and give a systematic convergence analysis. The reviews agree on the technical contributions, but relatively conservative on originality. In addition, the numerical experiments against batch gradient and accelerated gradient methods are not very convincing, since the state-of-arts for solving considered problems are randomized incremental or coordinate descent methods and their accelerated variants.